# Classification Systems of Endometrial Cancer: A Comparative Study about Old and New

**DOI:** 10.3390/diagnostics12010033

**Published:** 2021-12-24

**Authors:** Camelia Alexandra Coada, Giulia Dondi, Gloria Ravegnini, Antonio De Leo, Donatella Santini, Eugenia De Crescenzo, Marco Tesei, Alessandro Bovicelli, Susanna Giunchi, Ada Dormi, Marco Di Stanislao, Alessio G. Morganti, Dario De Biase, Pierandrea De Iaco, Anna Myriam Perrone

**Affiliations:** 1Center for Applied Biomedical Research, Alma Mater Studiorum-University of Bologna, 40138 Bologna, Italy; camelia.coada@unibo.it; 2Division of Oncologic Gynecology, IRCCS Azienda Ospedaliero-Universitaria di Bologna, 40138 Bologna, Italy; giulia.dondi@aosp.bo.it (G.D.); eugeniadecrescenzo@gmail.com (E.D.C.); marco.tesei@aosp.bo.it (M.T.); abovicelli@gmail.com (A.B.); susanna.giunchi@aosp.bo.it (S.G.); marco.distanislao@studio.unibo.it (M.D.S.); pierandrea.deiaco@unibo.it (P.D.I.); 3Centro di Studio e Ricerca delle Neoplasie Ginecologiche (CSR), University of Bologna, 40138 Bologna, Italy; antonio.deleo@unibo.it (A.D.L.); donatella.santini@aosp.bo.it (D.S.); alessio.morganti2@unibo.it (A.G.M.); dario.debiase@unibo.it (D.D.B.); 4Department of Pharmacy and Biotechnology (FABIT), University of Bologna, 40126 Bologna, Italy; gloria.ravegnini2@unibo.it; 5Department of Experimental, Diagnostic and Specialty Medicine, University of Bologna, 40138 Bologna, Italy; 6Pathology Unit, IRCCS Azienda Ospedaliero-Universitaria di Bologna, 40138 Bologna, Italy; 7Department of Medical and Surgical Sciences (DIMEC), University of Bologna, 40138 Bologna, Italy; ada.dormi@unibo.it; 8Radiation Oncology, IRCCS Azienda Ospedaliero-Universitaria di Bologna, 40138 Bologna, Italy

**Keywords:** endometrial cancer, TCGA, prognosis, ProMisE, ESMO risk

## Abstract

Endometrial cancer is the most common gynecological malignancy of the female reproductive organs. Historically it was divided into type I and type II, until 2013 when the Cancer Genome Atlas molecular classification was proposed. Here, we applied the different classification types on our endometrial cancer patient cohort in order to identify the most predictive one. We enrolled 117 endometrial cancer patients available for the study and collected the following parameters: age, body mass index, stage, menopause, Lynch syndrome status, parity, hypertension, type of localization of the lesion at hysteroscopy, type of surgery and complications, and presence of metachronous or synchronous tumors. The tumors were classified according to the European Society for Medical Oncology, Proactive Molecular Risk Classifier for Endometrial Cancer, Post-Operative Radiation Therapy in Endometrial Carcinoma, and Cancer Genome Atlas classification schemes. Our data confirmed that European Society for Medical Oncology risk was the strongest predictor of prognosis in our cohort. The parameters correlated with poor prognosis were the histotype, FIGO stage, and grade. Our study cohort shows that risk stratification should be based on the integration of histologic, clinical, and molecular parameters.

## 1. Introduction

Endometrial cancer (EC) is the most common gynecological neoplasm and has traditionally been classified into type I estrogen-related tumors (about 80% of cases) and type II non-estrogen-related cancers. This dualism represents two different types of cancer, according to etiology, histology, and prognosis. Type I usually occurs on a background of obesity with hyperestrogenism and is characterized by a low-grade endometrioid histotype with a good prognosis. Type II, on the other hand, mainly arises from polyps or Endometrial Intraepithelial Carcinoma (EIC) in a context of endometrial atrophy, estrogen unrelated. This tumor is represented by serous, clear cell histotypes or carcinosarcomas with poor prognosis. However, it has become progressively evident that the two groups overlap, making the dualistic model insufficient to adequately represent the heterogeneity of the condition [1,2,3]. This classification was used until 2013, when the molecular classification was applied [4].

In parallel, several studies demonstrated the prognostic importance of various surgical and pathological parameters, including histological type, grade, stage, depth of myometrial invasion, vascular invasion, and cervical involvement [5,6]. All those parameters are considered in the European Society for Medical Oncology (ESMO) risk classification scheme. In this stratification, the role of the pathologist becomes crucial in predicting the prognosis and the need for postoperative adjuvant treatment. However, this stratification is often correlated with poor reproducibility among different pathologists, mainly linked to individual experience. 

In recent years, these issues have led to the abandonment of conventional hormonal dualism and the search for a genetic and molecular classification [7,8], such as the one reported by The Cancer Genome Atlas (TCGA) in 2013 [9], which recognizes a more heterogeneous disease based on four subgroups: 1. ultramuted—POLEmut ECs, harboring pathogenic mutations in the *POLE* gene; 2. hypermuted—mismatch repair deficient (MMRd) ECs, showing microsatellite instability (MSI); 3. copy-number low—a group with no specific molecular profile (NSMP); and 4. copy-number high—serous-like TP35 EC—with mutations in *TP53* (p53abn). Given the cost and often the lack of proper instrumentation in routine diagnostic laboratories to perform this full genomic characterization, the TCGA dataset was used to develop replacement assays able to replicate the classification with the aim of incorporating it into clinical practice as an indicator of prognosis [10]. Considering that the TCGA classification did not have the statistical power to be clinically relevant, having resulted from an unsupervised clustering of genomic aberrations from a small and heterogeneous cohort, two groups, Vancouver and Post-Operative Radiation Therapy in Endometrial Carcinoma (PORTEC) [11], independently tried to confirm these results in larger cohorts with follow-up data. The Vancouver group coined the term ProMisE (the Proactive Molecular Risk Classifier for Endometrial Cancer) [12] and showed that tumors with POLEmut had the most favorable prognosis, those with p53 abn had the worst prognosis, and patients with p53 wild-type or MMRd EC had an intermediate prognosis. These groups are not only prognostically different but show distinctive responses to therapy which were not evaluable by the existing classifications [13,14,15,16,17,18]. Based on this, the WHO 2020 classification recommended the inclusion of molecular markers in clinical diagnosis; therefore, the new ESGO/ESTRO/ESP guidelines included them in the classes of risk as an adjunctive prognostic factor [19]. Even though these guidelines propose different therapeutic choices based on the classifications, they are not univocal; thus, prognosis of EC patients varies [20].

Given the existence of diverse classification schemes, the aim of this work was to analyze a cohort of EC patients available at our institution and to retrospectively apply the existing risk classifiers to evaluate and compare them. Clinical and pathological attributes were also analyzed in order to investigate associations with different tumoral features. Finally, a comparison between our cohort and the TCGA study cohort was carried out.

## 2. Materials and Methods

### 2.1. Patients

This retrospective observational multidisciplinary study included EC patients treated at the Division of Gynecologic Oncology, IRCCS Azienda Ospedaliero-Universitaria di Bologna (Bologna, Italy), between October 2010 and November 2019. The study was approved by the local research ethics committee CE-AVEC (Comitato Etico—Area Vasta Emilia Centro, registration n. 27/2019/Sper/AOUBo), and all the patients signed an informed consent form.

The inclusion criteria were (i) EC patients submitted to demolitive or conservative surgical treatment in our hospital, (ii) follow-up data available at least for two years (patients that relapsed or died within two years were also included), and (iii) EC specimen suitable for IHC and molecular analyses. The exclusion criterion was (i) chemotherapy or radiotherapy performed before surgery.

### 2.2. Surgical and Adjuvant Path of Our Study Cohort

Patients with a proven biopsy of EC collected during previous pre-operative work-up underwent minimally invasive surgery (MIS)—both robotic and laparoscopy—or laparotomic surgery based on the surgeon’s choice, according to the standard of care [19]. Surgical staging was performed according to ESGO-ESMO guidelines and included hysterectomy and bilateral salpingo-ovariectomy (BSO), with lymphadenectomy in high-grade cases and myometrial invasion greater than 50% at intraoperative frozen section (FS) examination. The sentinel node technique without lymphadenectomy was admitted in low-grade EC submitted to MIS. Peritoneal staging was performed in serous and clear cell carcinomas. All suspected lesions were removed and analyzed. Oophorectomy could be omitted in patients younger than 45 years and with myometrial invasion less than 50% confirmed at FS. Fertility sparing treatment was considered for all patients under 40 without myometrial invasion on Magnetic Resonance Imaging results and with a low-grade tumor [21]. In case of up-staging in the final pathology, patients received either a second surgical staging or radiation therapy based on the decisions of the multidisciplinary team. 

### 2.3. Data Collection

Patients’ data were retrieved from clinical, surgical, and pathologic records reported in a comprehensive clinicopathologic database. Clinical data included age, Body Mass Index (BMI), menopausal status, contraceptive use, hormonal replacement therapy (HRT), parity, comorbidities, metformin intake, personal and family cancer history, genetic assessment for Lynch Syndrome, surgical approach, staging and surgical complications (according to the Clavien–Dindo classification), adjuvant treatments, and follow-up data regarding recurrence and death.

Histology slides and all histopathologic parameters were reviewed by two expert pathologists (D.S., A.D.L.) according to the International Society of Gynecological Pathologists (ISGyP) [22]. Tumors were classified according to standard morphologic criteria following the World Health Organization classification of tumors [23], and the grade was evaluated using the International Federation of Gynecology and Obstetrics (FIGO) criteria [24]. Patients were also divided into risk groups according to the ESMO guidelines (low, intermediate, intermediate–high, and high) [5] and according to the PORTEC classification (low, intermediate, and high) [11].

The depth of myometrial invasion was recorded in all cases as a percentage of myometrial thickness. The pattern of myometrial invasion was reported, specifying whether it presented as microcystic, elongated, and fragmented (MELF) [25] and/or as single invasive cells or small groups of cells (tumor budding) [26]. Characteristics of the MELF pattern include the presence of invasive small, dilated glands lined by cuboidal or flattened cells with eosinophilic cytoplasm and with slit-like appearance. Tumor budding was defined as invasive single/small groups of cells without the formation of defined structures frequently lying in an edematous or myxoid background. A tumor budding was defined as a cluster of 1–4 tumor cells detached from the cohesive tumor part.

Lymph-vascular space invasion (LVSI) was defined by the presence of tumor fragments within endothelial-lined vascular/lymphatic spaces outside the immediate invasive border [27]. Intratumoral LVSI foci were not considered. A semi-quantitative three-tiered scoring system was applied: no LVSI, focal (a single focus of LVSI recognized around the tumor), or substantial (diffuse or multifocal LVSI around the tumor) [27,28].

Heterogeneity was defined as a tumor having two or more clearly separate morphological patterns, with each constituting at least 10% of the tumor [29]. Margins were defined as infiltrating/mixed or as pushing.

### 2.4. Assignment of the TCGA Molecular Classification

To assess the TCGA molecular classification, all cases were firstly evaluated for pathogenic *POLE* mutations. Diagnostic interpretation of *POLE* mutations was performed based on guidelines reported by Leon-Castillo et al. [15,30]. Then, MMR protein (*MLH1*, *MSH2*, *MSH6*, *PMS2*) expression was evaluated by immunohistochemical (IHC) assessment in order to identify MMR-deficient tumors. P53 status was also assessed by IHC; specifically, p53 was considered altered/abnormal if ≥50% of the tumor cells showed strong positive nuclear staining, or when areas (subclones) consisting of ≥50% positive tumor cells were present. Cases with no nuclear staining observed were further sequenced for *TP53* mutations. Finally, tumors with normal p53 and MMR expression by IHC, with no *POLE* alterations, were defined as NSMP tumors. 

To evaluate *POLE* and *TP53* mutations, genomic DNA was extracted from formalin-fixed, paraffin-embedded (FFPE) tumor tissue using the QIAamp DNA Micro Kit (Qiagen, Hilden, Germany) according to the manufacturer’s protocol. The two genes were investigated through a customized panel of genomic regions analyzing exons 9 to 14 for *POLE* and exons 4 to 9 for *TP53*, then sequenced using the Gene Studio S5 sequencer (ThermoFisher Scientific), according to the manufacturer’s instruction as previously reported [31]. 

In case of an abnormal or uncertain MMR IHC result, MSI analysis was carried out by PCR reaction using the CC-MSI kit (AB Analitica, Padova, Italy), which allows the analysis of 10 markers (BAT25, BAT26, D2S123, D5S346, D17S250, NR21, NR24, BAT40, TGFbRII, and D18S58). The fluorescent amplified PCR products were analyzed by capillary gel electrophoresis on an ABI 3730XL DNA Analyzer (Thermo Fisher Scientific, Waltham, MA, USA), using GeneMapper software, version 4.0 (Thermo Fisher Scientific, Waltham, MA, USA).

### 2.5. TCGA Study Cohort

MAF files for the TCGA-UCEC project were downloaded and explored using the R (Bioconductor, version 4.1.1) packages TCGAbiolinks (version 2.22.2) and maftools (version 2.10). The selection of pathogenic mutations of the TP53 gene was made considering the pathogenicity prediction by both the PolyPhen-2 (version 2.2.2) and SIFT (version 5.2.2) scoring systems. UCEC curated molecular subtypes derived from the TCGA marker paper were retrieved from synapse through TCGAbiolinks (version 2.22.2). For clinical variables, the progression-free interval (PFI) and overall survival (OS) were used as outcome variables, as recommended by the PanCanAtlas Publications NCI Genomic Data Commons guidelines [32]. The NSMP EC subgroup was selected by the exclusion of POLE, MMRd, and TP53 mutated tumors.

### 2.6. Statistical Analysis

Statistical analysis was performed using SPSS for Windows, version 20 (SPSS Inc., Chicago, IL, USA) and in RStudio (Bioconductor, version 4.1.1) [33]. Quantitative data are expressed as the mean ± SD (Standard Deviation), while qualitative data are expressed as the frequency and percentage. Comparisons between groups were realized using Student’s *t*-test and the Mann–Whitney, ANOVA, Chi-square, and Fisher tests, as appropriate. A *p*-value of <0.05 was considered significant. Missing data are presented as *NA* (not available) in the results tables but were not included in the statistical analysis. Kaplan–Meier analysis was used to build the survival and recurrence curves, while the log-rank test was used to calculate the significance. For overall survival, all deaths, irrespective of cause, were considered an event, while for recurrence-free survival (RFS), all recurrences (local, regional, and distant) were considered an event.

## 3. Results

### 3.1. Bologna Study Cohort

Table 1 reports the patient, surgery, and molecular characteristics of our cohort included in this study (Bologna study cohort). The same group of qualified gynecological pathologists carried out all histological analyses.

The endometrioid histotype was the most frequent among the analyzed EC patients, and low-grade (G1–2) tumors were predominant (*n* = 64, 54.7%). The revised 2009 FIGO stages for ECs at histological diagnosis were as follows: Stage I (*n* = 80, 68.3%) [IA *n* = 63, 53.8%; IB *n* = 17, 14.5%], Stage II (*n* = 5, 4.3%), and Stage III (*n* = 32, 27.4%) [IIIA *n* = 7, 6.0%; IIIB *n* = 2, 1.7%; IIIC *n* = 23, 19.7%].

Among our study population, 14 out of 117 patients (11.7%) had a disease recurrence. The recurrence rate was analyzed according to the different classification schemes available for ECs (i.e., ESMO, TCGA, ProMisE, and PORTEC) as shown in Table 1. Significant correlations were observed with respect to ESMO risk (*p* = 0.001). The ESMO classification showed that, among recurrent ECs, 92.9% were classified at high risk, whereas within the non-recurrent ECs, only 46% were at high risk. According to the PORTEC classification scheme, out of the recurrent ECs, 71.4% were classified as high risk vs. 14.3% at low risk, whereas within the non-recurrent ECs, 37.9% were at high risk vs. 29.1% at low risk.

Pathologic features were analyzed taking into consideration the presence of disease recurrence, and the results are summarized in Table 2.

Histology was significantly different between the two groups, with the endometroid type being more common in the non-recurrent ECs (70.9%), whereas carcinosarcoma was more frequent in recurrent ECs (35.7%), *p* = 0.014. FIGO stage was significantly associated with recurrence; in particular, the majority of recurrent ECs were classified as stage III (*n* = 8, 57.1%) [IIIA *n* = 1, 7.1%; IIIB *n* = 1, 7.1%; IIIC *n* = 6, 2.9%], but this was not the case in non-recurrent ECs (*n* = 24, 23.3%) [IIIA *n* = 6, 5.8%; IIIB *n* = 1, 1%; IIIC *n* = 17, 16.5%]; on the contrary, *n* = 76 (73.8%) of non-recurrent ECs were in stage I, versus 28.5% among recurrent ECs, *p* = 0.002. Similarly, grade was significantly associated with recurrence (low grade (G1–2): 28.6% vs. 58.3%; high grade (G3): 71.4% vs. 41.7%, in recurrent and non-recurrent ECs, respectively; *p* = 0.045). Lymph node metastases at diagnosis were more common in recurrent ECs (50% vs. 15.5%, *p* = 0.01). LVI was mainly found in patients who later developed a recurrence (57.1% vs. 27.2%, *p* = 0.015). The same parameters were also investigated with regard to time to progression (PFS) by log-rank test. Histology (*p* = 0.03), FIGO stage (*p* = 0.01), ESMO risk classification (*p* = 0.01), grade (*p* = 0.036), lymph node metastases (*p* = 0.014), LVI (*p* = 0.029), and margins (*p* = 0.043) were significantly associated with PFS (Figure 1).

None of the other analyzed parameters were significantly associated with recurrence.

Regarding overall survival (OS), significant associations were identified with FIGO stage (*p* = 0.001) and lymph node metastases (*p* = 0.015). The presence of recurrence was significantly correlated with shorter OS (*p* < 0.001), as shown in Figure 2.

Patients’ characteristics were then analyzed taking into consideration the TCGA classification scheme (Table 3 and Table 4).

The study cohort was categorized into POLE (*n* = 8, 6.8%), MMRd (*n* = 34, 29.1%), p53 (*n* = 30, 25.6%), and NSMP (*n* = 45, 38%). The POLE group included patients with a lower mean age, whereas BMI was significantly higher in the NSMP group (*p* = 0.009). Moreover, a higher percentage of NSMP patients presented diabetes (*p* = 0.025) and hypertension (*p* = 0.044); these results, taken together, clearly indicate that in NSMP ECs, there are associations with the pattern of metabolic syndrome.

The MELF pattern [34] of invasion was significantly correlated with TCGA classification (*p* = 0.005), with the POLE and MMRd subgroups presenting MELF in a higher number of cases (50% and 58.8% of ECs, respectively, versus 10% and 26.7% in p53 and NSMP Ecs).

By analyzing the effect of tumor budding within each of those molecular subgroups [35], we observed a significant difference between TCGA subgroups (*p* = 0.005). In particular, tumor budding was found in 75% and 50% of POLE and MMRd ECs, respectively, but only in 26.7% and 22.2% in p53 and NSMP EC cases, respectively. 

Heterogeneity was less common in MMRd and NSMP ECs compared to the other two subgroups (*p* = 0.031); interestingly, heterogeneity in NSMP cases was absent in about 76% of the cases.

### 3.2. TCGA Study Cohort

A total of 548 EC cases were retrieved from the TCGA database with available clinical, pathological, and surgical data. Out of all these ECs, 124 presented recurrence (22.8%). Regarding the histotype, endometrioid was the most frequent among the EC patients analyzed (75%), and high-grade tumors were predominant, accounting for 58.9% of ECs. The histology, stage, and grade were significantly different when comparing recurrent vs. non recurrent ECs; specifically, serous and grade 3 endometrioid endometrial carcinoma were the most common among the recurrent cases (*p* < 0.001, for both), while non-recurrent ECs were most frequently found in stage I (Table 5).

The molecular classification was also analyzed. According to the TCGA assessment, 65 cases were POLE mutated (17.3%), 110 were MMRd (29.3%), 54 were p53 mutated (14.4%), and 146 were NSMP ECs (38.9%). Even in this cohort, patients in the POLE group were significantly younger (*p* < 0.001), and patients in the NSMP group had a higher BMI compared with the other TCGA groups but without attaining statistical significance (Table 6).

## 4. Discussion

Over the last decades, several risk stratification schemes have been proposed in EC to better define the heterogeneity that characterizes this tumor type [6,15,36,37]. However, the existence of diverse classification schemes often makes it difficult to select the most appropriate one and causes additional clinical issues in terms of patient management.

The aim of this work was to analyze a cohort of EC patients available at our institution and to retrospectively apply the existing risk classifiers to evaluate and compare them. 

We analyzed a cohort of 117 EC patients, among which 11% were recurrent. The recurrence rate in the Bologna cohort was lower than that in the TCGA cohort [13]. However, when looking at this difference, we have to keep in mind that the duration of the follow-up period was different. Indeed, at the time of analysis, our cohort included patients with a follow-up period of up to 24 months, whereas in the TCGA cohort, this period was up to 8.6 years. Thus, it is reasonable to suppose that over time, additional patients of our cohort will experience recurrence, reaching a similar rate.

Looking at the Bologna cohort, the ESMO risk classification, comparing recurrent vs. non-recurrent ECs, appeared to be the best in predicting the patient prognosis. The PORTEC classification scheme was marginally statistically significant in the stratification of recurrent and non-recurrent patients (*p* = 0.057). On the contrary, a non-significant difference was observed when using the TCGA classification scheme. 

When we grouped patients according to the TGCA criteria, we observed that NSMP carcinomas showed more features associated with metabolic syndrome, including diabetes, obesity, and hypertension. This is in line with previous studies, and it may support estrogen-driven pathogenesis in this group of tumors [38].

The Bologna cohort was fully characterized from the clinico-pathological and molecular point of views, allowing us to investigate novel potential associations, including those with the tumor microenvironment characteristics and TCGA classification stratification. In our opinion, this is of particular interest because it may pave the way for further investigations in this, so-far under-considered, topic.

For example, in the last years, the importance of tumor budding as a prognostic marker has been reported in many solid tumors, including lung, pancreas, esophagus, and colorectal cancer [35,39,40]. In EC, the role of tumor budding has been poorly investigated, particularly when taking into account the TCGA classification [35]. Recently, Rau et al. showed that, in a cohort of 255 ECs, the presence of tumor budding in 26.3% of the cases was independently associated with a worse prognosis [35]. However, when the TCGA classification was used, tumor budding lost its prognostic value. In our study population, tumor budding was found in 35% of cases. When stratifying EC patients based on the TCGA scheme, the POLE and MMRd groups showed the highest numbers of cases with tumor budding. This is not in agreement with the report by Rau and collaborators, who found more than 50% of ECs with tumor budding in the NSMP group. This could be, at least in part, explained by the smaller sample size of our cohort.

Unfortunately, we were not able to investigate the same parameters in the TCGA cohort due to the fact that most of the clinical data were missing. However, this makes our result even more interesting, shedding light on a new aspect that deserves to be deepened in further independent and larger cohorts of patients.

The strength of this study is represented by the well-selected cohort comprising patients reviewed by pathologists who are dedicated to the diagnosis of gynecological cancers. The limitations of the study are represented by the retrospective data and the short follow-up time in the Bologna cohort; regarding the TCGA cohort, clinical data at diagnosis were not available and we could not perform a complete comparison.

In the last decade, constant efforts have been made to delineate an optimal classification in EC with high reproducibility in risk stratification to drive treatment choice, surgery, and surveillance; however, as suggested also by our analysis, additional efforts are needed to satisfy this yet-unmet clinical need.

Our findings clearly underline the central role of the histological characteristics in EC risk stratification. The molecular data should be considered from the perspective of an additional risk classification, and we believe that they must be integrated with the pathological data. This aspect also suggests that EC patients should be treated in referral centers where multi-expertise groups are able to accurately manage the path of care from surgical, pathological, and molecular points of view.

## 5. Conclusions

In conclusion, we tested different classifications in order to choose the best among the existing ones. Our data confirm that ESMO risk is more strongly related to patient prognosis than the new molecular classification. Our results suggest that a pathological examination performed by an expert pathologist still has a crucial role in the risk stratification of ECs. Molecular data provide important additional information that should be combined with the pathological data.

## Figures and Tables

**Figure 1 diagnostics-12-00033-f001:**
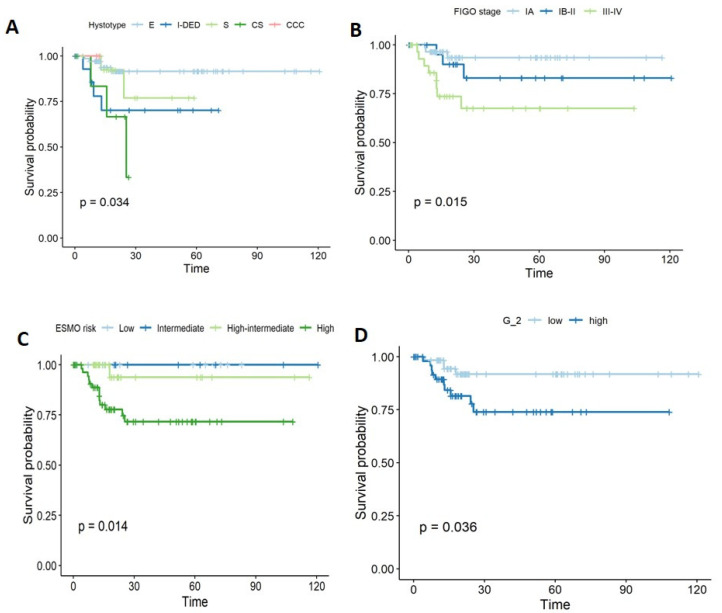
Kaplan–Meier curves showing the parameters associated with recurrence and (**A**) hystotype, (**B**) FIGO Stage, (**C**) ESMO risk, and (**D**) grade.

**Figure 2 diagnostics-12-00033-f002:**
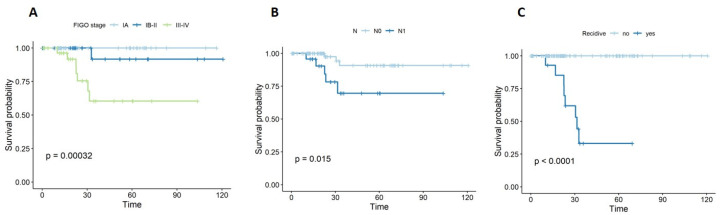
Kaplan–Meier curves for OS with respect to (**A**) stage, (**B**) lymph node metastasis, and (**C**) recurrence.

**Table 1 diagnostics-12-00033-t001:** Bologna study cohort: patient, surgery, and molecular characteristics of recurrent and non-recurrent EC patients.

KERRYPNX	All Cases *n* = 117	Recurrent ECs *n* = 14	Non-Recurrent ECs *n* = 103	*p* Value
Mean age, years (± SD)	62.8 ± 10.7	66.7 ±7.4	62.4 ± 11	0.155
Mean BMI (± SD)	27.7 ± 6.7	29 ± 7.9	27.4 ± 6.6	0.431
Lynch				
Yes	4 (3.4%)	0 (0%)	4 (3.9%)	
No	105 (89.7%)	14 (100%)	91 (88.3%)	1.000
N/A	8 (6.8%)	0 (0%)	8 (7.8%)	
Menopause				
Yes	101 (86.3%)	14 (100%)	87 (84.5%)	0.211
No	16 (3.7%)	0 (0%)	16 (15.5%)	
Cause of menopause				
Spontaneous	97 (82.9%)	13 (92.9%)	84 (81.6%)	0.459
Iatrogenic	20 (17.1%)	1 (7.1%)	19 (18.4%)	
HRT				
Yes	16 (13.7%)	3 (21.4%)	13 (12.7%)	0.416
No	97 (82.9%)	11 (78.6%)	86 (83.4%)	
NA	4 (3.4%)	0 (0%)	4 (3.9%)	
Parity				
Nulliparous	25 (21.4%)	3 (21.4%)	22 (21.4%)	1.000
Parous	92 (78.6%)	11 (78.6%)	81 (78.6%)	
ART				
Yes	3 (2.6%)	0 (0%)	3 (3%)	1
No	112 (97.4%)	14 (100%)	98 (97%)	
Hypertension				0.811
Yes	62 (53%)	7 (50%)	55 (53.4%)	
No	55 (47%)	7 (50%)	48 (46.6%)	
Diabetes				0.498
Yes	15 (12.8%)	1 (7.2%)	14 (13.6%)	
No	102 (87.2%)	13 (92.8%)	89 (86.4%)	
Metformin use				0.362
Yes	13 (11.1%)	0 (0%)	13 (12.6%)	
No	103 (88%)	14 (100%)	89 (86.4%)	
NA	1 (0.9%)	0 (0%)	1 (1%)	
Personal cancer history				
Yes	8 (6.8%)	1 (7.1%)	7 (6.8%)	1.000
No	109 (93.2%)	13 (92.9%)	96 (93.2%)	
Hysteroscopic localization of EC				1.000
Focal	31 (26.5%)	4 (28.6%)	27 (26.3%)	
Multifocal	39 (33.3%)	6 (42.8%)	33 (32%)	
NA	47 (40.2%)	4 (28.6%)	43 (41.7%)	
Aletti score				
≤3	78 (66.7%)	8 (57.1%)	70 (68%)	
4-7	36 (30.8%)	6 (42.9%)	30 (29.1%)	0.54
≥8	2 (1.7%)	0 (%)	2 (1.9%)	
NA	1 (0.9%)	0 (%)	1 (1%)	
Surgical approach				
Minimally invasive	62 (53%)	5 (35.7%)	57 (55.3%)	0.254
Laparotomy	55 (47%)	9 (64.3%)	46 (44.7%)	
Surgical Complications < >(Clavien–Dindo classification)				
No complications	100 (85.5%)	13 (92.9%)	87 (84.5%)	
Grade I	2 (1.7%)	0 (0%)	2 (1.9%)	
Grade II	12 (10.3%)	1 (7.1%)	11 (10.7%)	0.818
Grade III	3 (2.6%)	0 (0%)	3 (2.9%)	
Grade IV	0 (0%)	0 (0%)	0 (0%)	
Adjuvant therapies				
Yes	82 (70.1%)	13 (92.9%)	69 (67%)	0.062
No	35 (29.9%)	1 (7.1%)	34 (33%)	
ESMO risk group				**0.009**
Low	15 (12.8%)	0 (0%)	15 (14.6%)	
Intermediate	8 (6.9%)	0 (0%)	8 (7.8%)	
High–intermediate	35 (29.9%)	1 (7.1%)	34 (33%)	
High	59 (50.4%)	13 (92.9%)	46 (44.6%)	
TCGA classification				0.351
POLE	8 (6.8%)	0 (0%)	8 (7.8%)	
MMRd	34 (29%)	3 (21.4%)	31 (30%)	
p53	30 (25.7%)	6 (42.9%)	24 (23.4%)	
NSMP	45 (38.5%)	5 (35.7%)	40 (38.8%)	
ProMisE classification				0.371
POLE	6 (5.1%)	0 (0%)	6 (5.8%)	
MMRd	36 (30.7%)	3 (21.4%)	33 (32%)	
p53	30 (25.7%)	6 (42.9%)	24 (23.4%)	
NSMP	45 (38.5%)	5 (35.7%)	40 (38.8%)	
PORTEC risk group				**0.057**
Low	32 (27.4%)	2 (14.3%)	30 (29.1%)	
Intermediate	36 (30.7%)	2 (14.3%)	34 (33%)	
High	49 (41.9%)	10 (71.4%)	39 (37.9%)	
Lymphadenectomy				**0.038**
Yes	92 (78.6%)	14 (100%)	78 (75.7%)	
No	25 (21.4%)	0 (0%)	25 (24.3%)	
Synchronous EC-OC				0.593
Yes	8 (6.8%)	0 (0%)	8 (7.8%)	
No	104 (88.9%)	14 (100%)	90 (87.4%)	
NA	5 (4.3%)	0 (0%)	5 (4.8%)	

ART: assisted reproductive technology; BMI: body mass index; EC: endometrial cancer; EC-OC: endometrial cancer–ovarian cancer; ESMO: European Society for Medical Oncology; HRT: hormone replacement therapy; MMRd: mismatch repair deficient; NA: not available; NSMP: no specific molecular profile; PORTEC: Post-Operative Radiation Therapy in Endometrial Carcinoma; ProMisE: Proactive Molecular Risk Classifier for Endometrial Cancer; TCGA: The Cancer Genome Atlas. Bold highlights the statistical significance.

**Table 2 diagnostics-12-00033-t002:** Bologna study cohort: pathologic characteristics of recurrent and non-recurrent EC patients.

	All Cases *n* = 117	Recurrent ECs *n* = 14	Non-Recurrent ECs *n* = 103	*p* Value
Histology				**0.017**
Endometrioid	78 (66.7%)	5 (35.7%)	73 (70.9%)	
Indifferentiated/dedifferentiated	15 (12.8%)	4 (28.6%)	11 (10.7%)	
Serous	15 (12.8%)	2 (14.3%)	13 (12.6%)	
Carcinosarcoma	7 (6%)	3 (21.4%)	4 (3.9%)	
Clear Cell	2 (1.7%)	0 (0%)	2 (1.9%)	
FIGO Stage				**0.012**
IA	63 (53.8%)	3 (21.4%)	60 (58.3%)	
IB	17 (14.5%)	1 (7.1%)	16 (15.5%)	
II	5 (4.3%)	2 (14.4%)	3 (2.9%)	
IIIA	7 (6%)	1 (7.1%)	6 (5.8%)	
IIIB	2 (1.7%)	1 (7.1%)	1 (1%)	
IIIC	23 (19.7%)	6 (42.9%)	17 (16.5%)	
Grade				**0.047**
Low grade (G1–G2)	64 (54.7%)	4 (28.6%)	60 (58.3%)	
High grade (G3)	53 (45.3%)	10 (71.4%)	43 (41.7%)	
Lymph node metastasis				**0.010**
No	86 (73.5%)	7 (50%)	79 (76.7%)	
Yes	23 (19.7%)	7 (50%)	16 (15.5%)	
NA	8 (6.8%)	0 (0%)	8 (7.8%)	
MELF				0.769
Absent	75 (64.1%)	10 (71.4%)	65 (63.1%)	
Present	39 (33.3%)	4 (28.6%)	35 (33.9%)	
NA	3 (2.6%)	0 (0%)	3 (3%)	
Tumor budding				0.373
Absent	73 (62.4%)	11 (78.6%)	62 (60.1%)	
Present	41 (35%)	3 (21.4%)	38 (36.9%)	
N/A	3 (2.6%)	0 (0%)	3 (3%)	
LVI				**0.015**
Absent	81 (69.2%)	6 (42.9%)	75 (72.8%)	
Present	36 (30.8%)	8 (57.1%)	28 (27.2%)	
Myometrial invasion				0.596
<50%	85 (72.6%)	11 (78.6%)	74 (71.8%)	
>50%	32 (27.4%)	3 (21.4%)	29 (28.2%)	
Heterogeneity				0.276
Absent	72 (61.5%)	7 (50%)	65 (63.1%)	
Present	42 (35.9%)	7 (50%)	35 (34%)	
NA	3 (2.6%)	0 (0%)	3 (2.9%)	
Margins				**0.037**
Infiltrating/Mixed	86 (73.5%)	14 (100%)	72 (69.9%)	
Pushing	25 (21.4%)	0 (0%)	25 (24.3%)	
NA	6 (5.1%)	0 (0%)	6 (5.8%)	

FIGO: International Federation of Gynaecology and Obstetrics; G: grade; LVI: lymph-vascular space invasion; MELF: microcystic, elongated, and fragmented; NA: not available. Bold highlights the statistical significance.

**Table 3 diagnostics-12-00033-t003:** Bologna study cohort: patient and surgery characteristics according to TCGA classification.

	All Cases *n* = 117	POLE ECs *n* = 8	MMRd ECs *n* = 34	P53 ECs *n* = 30	NSMP ECs *n* = 45	*p* Value
Mean age, years (± SD)	62.8 ± 10.7	59.6 ± 13	63.9 ±10.5	65.6 ± 9.4	60.9 ±11.1	0.207
Mean BMI (± SD)	27.7 ± 6.7	25.8 ± 4.7	26.7 ± 6.5	25.2 ± 3.9	30.2 ± 7.9	**0.009**
HRT						
Yes	16 (13.7%)	2 (25%)	5 (14.7%)	5 (16.7%)	4 (8.9%)	0.602
No	97 (82.9%)	6 (75%)	28 (82.4%)	24 (80%)	39 (86.7%)	
NA	4 (3.4%)	0 (%)	1 (2.9%)	1 (3.3%)	2 (4.4%)	
Hypertension						**0.044**
Yes	62 (53%)	1 (12.5%)	18 (52.9%)	14 (46.7%)	29 (64.4%)	
No	55 (47%)	7 (87.5%)	16 (47.1%)	16 (53.3%)	16 (35.6%)	
Diabetes						**0.025**
Yes	15 (12.8%)	1 (12.5%)	2 (5.9%)	1 (3.3%)	11 (24.4%)	
No	102 (87.2%)	7 (87.5%)	32 (94.1%)	29 (96.7%)	34 (75.6%)	
Metformin use						0.101
Yes	13 (11.1%)	1 (12.5%)	2 (5.9%)	1 (3.3%)	9 (20%)	
No	103 (88%)	7 (87.5%)	32 (94.1%)	28 (93.4%)	36 (80%)	
NA	1 (0.9%)	0 (0%)	0 (0%)	1 (3.3%)	0 (0%)	
Hysteroscopic localization of EC						0.581
Focal	31 (26.5%)	2 (%)	12 (%)	5 (16.7%)	12 (26.7%)	
Multifocal	39 (33.3%)	4 (%)	10 (%)	10 (33.3%)	15 (33.3%)	
NA	47 (40.2%)	2 (%)	12 (%)	15 (50%)	18 (40%)	
ESMO risk group						0.164
Low	15 (12.8%)	1 (12.5%)	6 (17.6%)	0 (0%)	8 (17.8%)	
Intermediate	8 (6.9%)	0 (0%)	4 (11.8%)	1 (3.3%)	3 (6.7%)	
High–intermediate	35 (29.9%)	2 (25%)	7 (20.6%)	2 (6.7%)	24 (53.3%)	
High	59 (50.4%)	5 (62.5%)	17 (50%)	27 (90%)	10 (22.2%)	
PORTEC risk group						0.236
Low	32 (27.4%)	8 (100%)	0 (0%)	0 (0%)	24 (53.3%)	
Intermediate	36 (30.7%)	0 (0%)	23 (67.6%)	0 (0%)	13 (28.9%)	
High	49 (41.9%)	0 (0%)	11 (32.4%)	30 (100%)	8 (17.8%)	
Lymphadenectomy						**0.01**
Yes	92 (78.6%)	8 (100%)	27 (79.4%)	28 (93.3%)	29 (64.4%)	
No	25 (21.4%)	0 (0%)	7 (20.6%)	2 (6.7%)	16 (35.6%)	
Synchronous EC-OC						0.281
Yes	8 (6.8%)	1 (12.5%)	0 (0%)	3 (10%)	4 (8.9%)	
No	104 (88.9%)	6 (75%)	33 (97.1%)	26 (86.7%)	39 (86.7%)	
NA	5 (4.3%)	1 (12.5%)	1 (2.9%)	1 (3.3%)	2 (4.4%)	

BMI: body mass index; EC-OC: endometrial cancer–ovarian cancer; ESMO: European Society for Medical Oncology; HRT: hormonal replacement therapy; MMRd: mismatch repair deficiency; NA: not available; NSMP: no specific molecular profile; PORTEC: Post-Operative Radiation Therapy in Endometrial Carcinoma; TCGA: The Cancer Genome Atlas. Bold highlights the statistical significance.

**Table 4 diagnostics-12-00033-t004:** Bologna study cohort: pathologic characteristics according to TCGA classification.

	All Cases *n* = 117	POLE ECs *n* = 8	MMRd ECs *n* = 34	P53 ECs *n* = 30	NSMP ECs *n* = 45	*p* Value
Histology						**<0.001**
Endometrioid	78 (66.7%)	7 (87.5%)	24 (70.6%)	5 (16.7%)	42 (93.3%)	
Undifferentiated/dedifferentiated	15 (12.8%)	1 (12.5%)	10 (29.4%)	1 (3.3%)	3 (6.7%)	
Serous	15 (12.8%)	0 (0%)	0 (0%)	15 (50%)	0 (0%)	
Carcinosarcoma	7 (6%)	0 (0%)	0 (0%)	7 (23.3%)	0 (0%)	
Clear Cell	2 (1.7%)	0 (0%)	0 (0%)	2 (6.7%)	0 (0%)	
FIGO Stage						**0.056**
IA	63 (53.8%)	4 (50%)	15 (44.2%)	12 (40%)	32 (71.2%)	
IB	17 (14.5%)	2 (25%)	9 (26.5%)	2 (6.7%)	4 (8.9%)	
II	5 (4.3%)	0 (0%)	1 (2.9%)	2 (6.7%)	2 (4.4%)	
IIIA	7 (6%)	1 (12.5%)	1 (2.9%)	3 (10%)	2 (4.4%)	
IIIB	2 (1.7%)	0 (0%)	2 (5.9%)	0 (0%)	0 (%)	
IIIC	23 (19.7%)	1 (12.5%)	6 (17.6%)	11 (36.6%)	5 (11.1%)	
Grade						**<0.001**
Low grade (G1–G2)	64 (54.7%)	4 (50%)	22 (64.7%)	1 (3.3%)	37 (82.2%)	
High grade (G3)	53 (45.3%)	4 (50%)	12 (35.3%)	29 (96.7%)	8 (17.8%)	
Lymph node metastasis						0.067
No	86 (73.5%)	7 (87.5%)	26 (76.5%)	18 (60%)	35 (77.8%)	
Yes	23 (19.7%)	1 (12.5%)	6 (17.6%)	11 (36.7%)	5 (11.1%)	
NA	8 (6.8%)	0 (0%)	2 (5.9%)	1 (3.3%)	5 (11.1%)	
MELF						**<0.001**
Absent	75 (64.1%)	4 (50%)	13 (38.3%)	27 (90%)	31 (68.9%)	
Present	39 (33.3%)	4 (50%)	20 (58.8%)	3 (10%)	12 (26.7%)	
NA	3 (2.6%)	0 (0%)	1 (2.9%)	0 (%)	2 (4.4%)	
Tumor budding						**0.005**
Absent	73 (62.4%)	2 (25%)	16 (47.1%)	22 (73.3%)	33 (73.4%)	
Present	41 (35%)	6 (75%)	17 (50%)	8 (26.7%)	10 (22.2%)	
NA	3 (2.6%)	0 (0%)	1 (2.9%)	0 (0%)	2 (4.4%)	
LVI						**0.050**
Absent	81 (69.2%)	6 (75%)	24 (70.6%)	15 (50%)	36 (80%)	
Present	36 (30.8%)	2 (25%)	10 (29.4%)	15 (50%)	9 (20%)	
Myometrial invasion						0.131
<50%	85 (72.6%)	7 (87.5%)	22 (64.7%)	20 (66.7%)	36 (80%)	
>50%	32 (27.4%)	1 (12.5%)	12 (35.3%)	10 (33.3%)	9 (20%)	
Heterogeneity						**0.031**
Absent	72 (61.5%)	4 (50%)	20 (58.9%)	14 (46.7%)	34 (75.6%)	
Present	42 (35.9%)	4 (50%)	13 (38.2%)	16 (53.3%)	9 (20%)	
NA	3 (2.6%)	0 (0%)	1 (2.9%)	0 (%)	2 (4.4%)	
Margins						0.679
Infiltrating/Mixed	86 (73.5%)	5 (62.5%)	27 (79.5%)	21 (70%)	33 (73.3%)	
Pushing	25 (21.4%)	3 (37.5%)	6 (17.6%)	7 (23.3%)	9 (20%)	
NA	6 (5.1%)	0 (0%)	1 (2.9%)	2 (6.7%)	3 (6.7%)	

G: grade; FIGO: International Federation of Gynaecology and Obstetrics; LVI: lymph-vascular space invasion; MELF: microcystic, elongated, and fragmented; MMRd: mismatch repair deficiency; NA: not available; NSMP: no specific molecular profile. Bold highlights the statistical significance.

**Table 5 diagnostics-12-00033-t005:** TCGA study cohort: patient and pathologic characteristics of recurrent and non-recurrent ECs.

	All Cases *n* = 548	Recurrent ECs *n* = 124	Non-Recurrent ECs *n* = 424	*p* Value
Mean age, years (± SD)	63.9 ± 11.1	64.6 ± 9.7	63.7 ± 11.5	0.37
Mean BMI (± SD)	33.8 ± 12.1	33.2 ± 8.4	34 ± 12.9	0.53
Surgical approach				
Minimally invasive	203(37)	55(44.4)	148(34.9)	**0.04**
Laparotomy	321(58.6)	63(50.8)	258(60.8)	
N/A	24(4.4)	6(4.8)	18(4.2)	
ProMisE classification				
POLE	65(17.3)	3(3.8)	62(20.9)	**<0.001**
MMRd	110(29.3)	24(30.8)	85(29)	
p53abn	54(14.4)	22(28.2)	32(10.8)	
NSMP	146(38.9)	29(37.2)	117(39.4)	
Histology				
Endometrioid	411(75)	74(59.7)	337(79.5)	**<0.001**
Mixed serous and endometrioid	22(4)	8(6.5)	14(3.3)	
Serous endometrial adenocarcinoma	115(21)	42(33.9)	73(17.2)	
FIGO Stage				
I	245(65.3)	37(47.4)	208(70)	**<0.001**
II	32(8.5)	3(3.8)	29(9.8)	
III	82(21.9)	26(33.3)	56(18.9)	
IV	16(4.3)	12(15.4)	4(1.3)	
Grade				
G1	99(18.1)	10(8.1)	89(21)	**<0.001**
G2	122(22.3)	21(16.9)	101(23.8)	
G3	327(58.9)	93(75)	234(55.2)	

BMI: body mass index; EC: endometrial cancer; FIGO: International Federation of Gynaecology and Obstetrics; G: grade; MMRd: mismatch repair deficiency; NA: not available; NSMP: no specific molecular profile; ProMisE: Proactive Molecular Risk Classifier for Endometrial Cancer; TCGA: The Cancer Genome Atlas. Bold highlights the statistical significance.

**Table 6 diagnostics-12-00033-t006:** TCGA study cohort: patient and pathologic characteristics of EC according to TCGA classification.

Characteristics	All Cases *n* = 373	POLE ECs *n* = 63	MMRd ECs *n* = 110	P53 ECs *n* = 54	NSMP ECs *n* = 146	*p* Value
Mean age, years (± SD)	62.8 ± 11.3	58.4 ± 12	63 ± 9.8	68.7 ± 8.7	62.4 ± 11.9	<0.001
Mean BMI (± SD)	33.8 ± 13.1	32.4 ± 25.2	33.3 ± 7.6	32.3 ± 10.7	35.5 ± 9.1	0.27
Histology						
Endometrioid	315(84)	62(95.4)	104(94.5)	19(35.2)	130(89)	**<0.001**
Mixed serous and endometroid	12(3.2)	1(1.5)	2(1.8)	6(11.1)	3(2.1)	
Serous	48(12.8)	2(3.1)	4(3.6)	29(53.7)	13(8.9)	
FIGO Stage						
I	245(65.3)	41(63.1)	79(71.8)	24(44.4)	101(69.2)	**0.02**
II	32(8.5)	7(10.8)	8(7.3)	5(9.3)	12(8.2)	
III	82(21.9)	15(23.1)	19(17.3)	17(31.5)	31(21.2)	
IV	16(4.3)	2(3.1)	4(3.6)	8(14.8)	2(1.4)	
Grade						
G1	91(24.3)	11(16.9)	27(24.5)	-	53(36.3)	**<0.001**
G2	103(27.5)	12(18.5)	32(29.1)	5(9.3)	54(37)	
G3	181(48.3)	42(64.6)	51(46.4)	49(90.7)	39(26.7)	

BMI: body mass index; EC: endometrial cancer; FIGO: International Federation of Gynaecology and Obstetrics; G: grade; MMRd: mismatch repair deficiency; NSMP: no specific molecular profile; TCGA: The Cancer Genome Atlas; SD: standard deviation. Bold highlights the statistical significance.

## Data Availability

The data presented in this study are available on request from the corresponding author.

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
