# Peer review of "Classification Systems of Endometrial Cancer: A Comparative Study about Old and New"

_diagnostics, 2021, doi:10.3390/diagnostics12010033_

Round 1
Reviewer 1 Report
It is a very interesting and well written article. Authors provide useful data regarding various classification systems of endometrial cancer. More specifically, they describe in detail molecular classification system which is based on Cancer Genome Atlas (TCGA). They present demographic, histological and molecular data of their study population, as well as their classification based on the described systems. They also investigated possible correlations with patients' prognosis. Overall, it is a well written article.
Author Response
Dear Reviewer,
Thank you for the time you dedicated to the revision of our paper and for your positive comments. We are glad that you have found our article of interest and worthy of publication. Your feedback is important for the quality of our work.
Reviewer 2 Report
The present work investigates the impact of molecular and clinicopathological parameters in a cohort of patients with endometrial cancer. The work is interesting for the accurate description of the characteristics of the cohort in particular the histopathological features. Overall, the study is well structured and the comparison of the study cohort with that of TCGA is interesting. This study focuses on the importance of multimodal integration in the classification of endometrial cancer.However, the paper requires revision of the English, particularly in the "discussion" section. In addition, some inconsistencies need to be resolved.
I report some notes that should be corrected by the authors:
1. in the abstract there are some errors: for example line 41-43: the ESMO risk is based on histopathological parameters so this sentence is not correct, if anything it should say the exact opposite that is that the histopathological parameters that are included in the ESMO stratification are solid parameters. It is misleading to talk about TILs in the abstract, also because it is not clear what the acronym refers to. I think the more correct message is that good risk stratification should be based on the integration of histologic, clinical, and molecular parameters.
2. lines 72-75: the authors have inverted the high copy-numbers that are p53 abnormal with the low copy-numbers that are NSMP.
3. reference 13 is wrong and misleading, it should refer to the main ProMiSe study: doi: 10.1002/cncr.30496, it should be corrected: "Tumors with POLE EDMs had the most favorable prognosis, and those with p53 abn the worst prognosis, and separation of the 2 middle survival curves (p53 wt and MMR-D) was observed."
4. line 138 correct: throughly
5. correct everywhere FIGO stage classification, it is not a classification but a staging: FIGO stage
6. line 264: in the study I would not include the TILs data, it seems to me that it is out of the objectives of the study: to compare the old with the new.
7. in general it is always better to refer to tumor budding, the acronym BUD is confusing.
9. to use TCGA surrogate classification instead of TCGA classification, in the text there are several errors with the letters reversed
10. the authors overlook an important finding: in NSMP carcinomas there are associations with the pattern of metabolic syndrome, this finding should be appropriately discussed
11. use in a consistent way MSI or MMRd, in the text is confusing to alternate the two acronyms, in the discussion in particular
12. line 327: years or months? Please correct
13. lines 340-341: I repeat that I think it is confusing to talk about TILs in this context.
14. the discussion should be reviewed and better articulated, what is clear from the study is that histological features may have an important role in risk stratification, in fact ESMO risk is based on these characteristics. The molecular data should be integrated in this vision to add further risk stratification: I believe that these are the most appropriate considerations for the study.
The study is interesting for this reason, i.e. a correct prognostic stratification cannot disregard an adequate histopathological examination.
It must be stressed that molecular features must be integrated, in the present series the non-significance should be explained: e.g. follow-up time, treatment, sample size.
15. conclusions are not very strong, review previous comment.
16. important references should be added: doi: 10.1200/JCO.20.00549; doi: 10.1007/s00404-021-06028-4; doi: 10.1158/1078-0432.CCR-15-2878; doi: 10.1016/j.ygyno.2020.01.008; doi: 10.1093/jnci/dju402.
17. one observation: there are few pathologists (only two) and many gynecologists in the study. In consideration of the importance of the histological parameters inferred in the study, please evaluate the authorships carefully.
Author Response
Dear Reviewer,
Thank you for your suggestions that will improve the quality of our paper. The required revisions are reported point by point below.
1.In the abstract there are some errors: for example line 41-43: the ESMO risk is based on histopathological parameters so this sentence is not correct, if anything it should say the exact opposite that is that the histopathological parameters that are included in the ESMO stratification are solid parameters. It is misleading to talk about TILs in the abstract, also because it is not clear what the acronym refers to. I think the more correct message is that good risk stratification should be based on the integration of histologic, clinical, and molecular parameters.
Thank you for the comments the sentence “ITIls and sTIls were correlated with progression-free survival” in line 43-44 was eliminated. The sentence at line 41-43 was changed according to your suggestions and the sentence “Our study cohort shows that risk stratification should be based on the integration of histologic, clinical, and molecular parameters” was added.
lines 72-75: the authors have inverted the high copy-numbers that are p53 abnormal with the low copy-numbers that are NSMP.
We corrected this error at line 82-83 and the sentence was modified as follow: “copy-number low, a group with no specific molecular profile (NSMP) and 4. copy-number high – serous like TP35 EC – with mutations in TP53 (p53abn)”.
- reference 13 is wrong and misleading, it should refer to the main ProMiSe study: doi: 10.1002/cncr.30496, it should be corrected: "Tumors with POLE EDMs had the most favorable prognosis, and those with p53 abn the worst prognosis, and separation of the 2 middle survival curves (p53 wt and MMR-D) was observed."
Thank you for the suggestion the citation was changed and the sentence in line 94-95 was replaced with the suggested one: “..and showed that tumors with POLEmut had the most favorable prognosis, those with p53 abn the worst prognosis, while patients with p53 wt or MMRd EC had an intermediate prognosis”.
- line 138 correct: thoroughly
In line 165 we eliminated thoroughly.
- correct everywhere FIGO stage classification, it is not a classification but a staging: FIGO stage
It was done, FIGO stage classification was replaced with FIGO stage in the text.
- line 264: in the study I would not include the TILs data, it seems to me that it is out of the objectives of the study: to compare the old with the new.
Thank you for the suggestions, the part of TILs data was deleted as required. In particular, the sentences in line 189-194, 321-324, 377-381, 630-643 and 842-843 were cancelled. The TILs data in tables (table 2 and table 4) were deleted. Figure 3 was removed. The bibliography was updated.
- in general it is always better to refer to tumor budding, the acronym BUD is confusing.
Thank for the suggestion BUD was replaced with tumor budding in the text and tables.
- to use TCGA surrogate classification instead of TCGA classification, in the text there are several errors with the letters reversed
Thank you again, we checked and corrected the type errors in the text and tables.
- the authors overlook an important finding: in NSMP carcinomas there are associations with the pattern of metabolic syndrome, this finding should be appropriately discussed
Thank you for your recommendation We agree with the reviewer and based on that we highlighted this finding.
We added a sentence (line 387-390) summarizing the results reported in table 3: “Moreover, a higher percentage of NSMP patients showed diabetes (p=0.025) and hypertension (p=0.044); these results, taken together, clearly indicate that in NSMP ECs there are associations with the pattern of metabolic syndrome”
We also included this point the discussion to underline the association in line 570-573: “When we grouped patients according to the TGCA criteria, we observed that NSMP carcinomas showed more features associated with metabolic syndrome including diabetes, obesity and hypertension. This is in line with previous studies and it may support the estrogen driven pathogenesis in this group of tumors”
- use in a consistent way MSI or MMRd, in the text is confusing to alternate the two acronyms, in the discussion in particular
Thank you, we used MMRd ECs instead of MSI ECs and all MSI has been correct in the text and tables.
- line 327: years or months? Please correct
Sorry for the misunderstanding, we corrected it. The follow up time was expressed in years.
- lines 340-341: I repeat that I think it is confusing to talk about TILs in this context
Thank you for your suggestion we removed all TILs data in text, tables and figures as suggested.
- the discussion should be reviewed and better articulated, what is clear from the study is that histological features may have an important role in risk stratification, in fact ESMO risk is based on these characteristics. The molecular data should be integrated in this vision to add further risk stratification: I believe that these are the most appropriate considerations for the study.
The study is interesting for this reason, i.e. a correct prognostic stratification cannot disregard an adequate histopathological examination.
It must be stressed that molecular features must be integrated, in the present series the non-significance should be explained: e.g. follow-up time, treatment, sample size.
Thank you for your recommendation, in line 830-839 we added the sentence “In the last decade, constant efforts have been done to delineate an optimal classification in EC with a high reproducibility in the risk stratification to drive treatment choice, surgery and surveillance; however, as suggested also by our analysis, additional efforts are needed to satisfy this yet unmet clinical need.
Our findings clearly underline the central role of the histological characteristics in EC risk stratification. The molecular data should be considered in the prospective of an additional risk classification and we believed that they must be integrated with the pathological data. This aspect also suggests that EC patients should be treated in referral centers where multi-expertise groups are able to accurately manage the path of care from surgical, pathological, molecular point of views.”
- conclusions are not very strong, review previous comment.
Thank you for the suggestion in line 837-840 we added the sentence “Our results suggest that the pathological examination performed by an expert pathologist still has a crucial role in the risk stratification of ECs. Molecular data provides important additional information that should be combined with the pathological data.”
- important references should be added: doi: 10.1200/JCO.20.00549; doi: 10.1007/s00404-021-06028-4; doi: 10.1158/1078-0432.CCR-15-2878; doi: 10.1016/j.ygyno.2020.01.008; doi: 10.1093/jnci/dju402.
The suggested references have been added.
- one observation: there are few pathologists (only two) and many gynecologists in the study. In consideration of the importance of the histological parameters inferred in the study, please evaluate the authorships carefully.
The authorship was re-check, and it was correct. Two pathologists (A.D.L. and D.S.) are involved. D.D.B. and C.A.C are affiliated with the University of Bologna and performed the molecular analysis.